# Fronto-Parietal Gray Matter Volume Loss Is Associated with Decreased Working Memory Performance in Adolescents with a First Episode of Psychosis

**DOI:** 10.3390/jcm10173929

**Published:** 2021-08-31

**Authors:** Marta Rapado-Castro, Mara Villar-Arenzana, Joost Janssen, David Fraguas, Igor Bombin, Josefina Castro-Fornieles, Maria Mayoral, Ana González-Pinto, Elena de la Serna, Mara Parellada, Dolores Moreno, Beatriz Paya, Montserrat Graell, Inmaculada Baeza, Christos Pantelis, Celso Arango

**Affiliations:** 1Department of Child and Adolescent Psychiatry, Institute of Psychiatry and Mental Health, Hospital General Universitario Gregorio Marañón, School of Medicine, Universidad Complutense, IiSGM, CIBERSAM, 28040 Madrid, Spain; joost.janssen@iisgm.com (J.J.); maria.mayoral@iisgm.com (M.M.); parellada@hggm.es (M.P.); lolamoreno@hggm.es (D.M.); carango@hggm.es (C.A.); 2Melbourne Neuropsychiatry Centre, Department of Psychiatry, The University of Melbourne and Melbourne Health, 161 Barry Street, Carlton South, VIC 3053, Australia; cpant@unimelb.edu.au; 3Department of Psychiatry, School of Medicine, Universidad Complutense, 28040 Madrid, Spain; maravi01@ucm.es; 4Institute of Psychiatry and Mental Health, Hospital Clínico San Carlos, IdISSC, CIBERSAM, School of Medicine, Universidad Complutense, 28040 Madrid, Spain; david.fraguas@iisgm.com; 5Centro de Investigación Biomédica en Red de Salud Mental (CIBERSAM), 33011 Oviedo, Spain; ibombin@reintegra-dca.es; 6Reintegra Foundation, 33011 Oviedo, Spain; 7Child Psychiatry and Psychology Department, 2017SGR881, Neurosciences Institute, Hospital Clinic de Barcelona, IDIBAPS (Institut d’Investigacions Biomèdiques August Pi Sunyer), CIBERSAM (Centro deInvestigación Biomédica en Red de Salud Mental), Department of Medicine University of Barcelona, 28036 Barcelona, Spain; jcastro@clinic.cat (J.C.-F.); ESERNA@clinic.cat (E.d.l.S.); IBAEZA@clinic.cat (I.B.); 8Hospital Universitario de Álava, BIOARABA, Centro de Investigación Biomédica en Red de Salud Mental, CIBERSAM, Kronikgune, EHU-UPV, 01009 Vitoria, Spain; anapinto@telefonica.net; 9Department of Psychiatry, Hospital Universitario del Sureste, 28500 Madrid, Spain; 10Child Psychiatry Unit, Hospital Universitario Marqués de Valdecilla, Centro de Investigación Biomédica en Red de Salud Mental, CIBERSAM, 39008 Santander, Spain; bpaya@humv.es; 11Psychiatry and Psychology Department, Hospital Infantil Universitario Niño Jesús, Centro de Investigación Biomédica en Red de Salud Mental, CIBERSAM, 28009 Madrid, Spain; montserratgraell1@gmail.com

**Keywords:** working memory, attention, executive function, early onset psychosis, brain volume, adolescence, gray matter, magnetic resonance imaging, MRI

## Abstract

Cognitive maturation during adolescence is modulated by brain maturation. However, it is unknown how these processes intertwine in early onset psychosis (EOP). Studies examining longitudinal brain changes and cognitive performance in psychosis lend support for an altered development of high-order cognitive functions, which parallels progressive gray matter (GM) loss over time, particularly in fronto-parietal brain regions. We aimed to assess this relationship in a subsample of 33 adolescents with first-episode EOP and 47 matched controls over 2 years. Backwards stepwise regression analyses were conducted to determine the association and predictive value of longitudinal brain changes over cognitive performance within each group. Fronto-parietal GM volume loss was positively associated with decreased working memory in adolescents with psychosis (frontal left (B = 0.096, *p* = 0.008); right (B = 0.089, *p* = 0.015); parietal left (B = 0.119, *p* = 0.007), right (B = 0.125, *p* = 0.015)) as a function of age. A particular decrease in frontal left GM volume best predicted a significant amount (22.28%) of the variance of decreased working memory performance over time, accounting for variance in age (14.9%). No such association was found in controls. Our results suggest that during adolescence, EOP individuals seem to follow an abnormal neurodevelopmental trajectory, in which fronto-parietal GM volume reduction is associated with the differential age-related working memory dysfunction in this group.

## 1. Introduction

The development of cognitive function plays a vital role in the ability of an individual to relate to the world. Altered cognitive functioning, particularly in high-order cognitive processes (i.e., sustained attention, working memory and executive function) has consistently been shown in early psychosis [1,2], leading to significant social, functional and vocational impairments and poor quality of life [3,4,5,6]. Cognitive maturation is largely influenced by the maturation of brain cortical structures. In particular, high-order cognitive processes have generally been associated with fronto-parietal brain regions, which appear to be the last regions to mature. Of particular importance is the prefrontal cortex, in which the connections with the parietal lobes are involved with cognitive abilities required for planning, decision making and complex problem solving, such as set-shifting, selective and sustained attention and working memory [7,8,9,10,11]. Reduced brain gray matter (GM) volume over time is characteristic in psychosis [12,13]. Our own clinical studies lend support for an altered development of higher brain cognitive functions [14], which parallels progressive GM loss in early onset psychosis (EOP) over time, particularly in fronto-parietal brain regions [15,16,17,18,19]. However, the relationship between these processes is not well understood.

Adolescence is a time of substantial brain and cognitive development. Gray matter changes/loss in the frontal, parietal, temporal and occipital lobes have been described as part of normal development [20], and these changes have shown to be accelerated in adolescent psychosis, particularly in early stages, for review see [21]. In this regard, the maturation of cognitive functions may coincide with synaptogenesis in the frontal and parietal cortex (i.e., synaptic pruning [22]) in adolescence and young adulthood, so that a disruption of those maturational processes may lead to the cognitive dysfunction observed in these individuals. However, little research has been done in this early onset psychosis (EOP) population. Our own previous studies in EOP have shown that cognitive impairment is present at the time of the first episode [23], and that cognitive development seems to be arrested at the 2 year follow-up, at least for some high-order functions, such as sustained attention or working memory [14]. Research has identified the development of high-order cognition to begin as early as 12 months of age [24,25], and exhibits a rapid development during childhood and adolescence [26]. Additionally, different aspects of higher cognitive functions are found to display different developmental trajectories [24]: Sustained attention, a basic underlying cognitive process required to complete any activity, is the first one to develop. While working memory and sequencing ability may be developed by the age of 15 to 19, strategic planning, problem solving and goal-directed behavior may have been achieved between the ages of 20 and 25 [24]. Thus, the development of high-order cognitive abilities is a continued process up until the late 20s, when brain development is considered to be complete [27]. However, the onset of psychosis may interfere with the developmental process [28]. In this regard, early neurodevelopmental abnormalities may underlie the differential acquisition of cognitive abilities in individuals with psychosis.

Previous studies investigating associations of GM volume and cognition in schizophrenia have been conducted mostly at a single point and in adult or young adult populations with inconsistent findings. Cognitive performance in these participants have been associated with whole-brain GM volumes [29,30,31,32,33,34], total brain volume [35], volumes of the frontal [36,37,38,39,40,41,42,43,44,45,46,47,48], parietal [41], temporal [40,46,49], cingulate [42], temporo-occipital [50], temporolimbic [51,52] or hippocampal [43,47] brain areas. Variability across cognitive and neuroimaging measurement strategies, antipsychotic treatment, patient heterogeneity, age of onset, duration of psychosis and duration of untreated psychosis may have contributed to inconsistent results in these studies. Two studies in particular have investigated the GM volume–cognition relationship in young adults with FEP [53,54]. A cross-sectional study investigating the relationship between brain volume and a combined measure of cognitive function (including measures of attention, working memory and verbal fluency such as the forward and backward digit span test from the Wechsler Memory Scale and the Controlled Oral Word Association Test, COWAT) suggest an association between GM volume in the frontal, parietal and temporal cortices, specifically inferior regions of the dorsolateral prefrontal cortex and cognitive performance in 122 FEP individuals (with mean age 28.9 (8.6) years) [53]. A longitudinal study investigating the association of GM volume and cognitive function (working memory, executive functioning and processing speed) over the early course of FEP found an association between GM volume decline over 80 months in the left parietal lobe, and worse executive function performance at baseline [54]. In the same study, an association between GM loss in the globus pallidus and left inferior parietal lobule, and a lower processing speed at baseline was found in a sample of 16 drug-naïve non-affective FEP participants (mean age 24.08 (7.21)) [54]. Finally, in a cross-sectional investigation in a mixed sample of 41 individuals with chronic schizophrenia and 4 FEP subjects, (mean age 40.49 (11.67)) significant correlations were found between generalized cognitive deficits (lower premorbid IQ, verbal memory, attention and processing speed) and GM reductions within left fronto-temporal regions [55]. Only a previous attempt in our group examined cognitive baseline predictors of 2 year GM volume reductions in first episodes of adolescent EOP, reporting a specific relationship between low IQ at baseline and reduced GM in bilateral frontal cortices in the subgroup of 34 participants with early onset schizophrenia (mean age 15.2 (1.7)) [19]. In addition, low performance in working memory at baseline was associated with decreased GM volume in the frontal lobe (bilaterally) in this same subsample [19]. To the best of our knowledge, no previous studies have explored this brain–cognition relationship longitudinally, neither in first-episode EOP nor in the context of a particularly sensitive developmental window such as adolescence.

The aim of this study is to explore the relationship between brain changes in frontal and parietal GM volume, and changes in high-order cognitive functions (particularly sustained attention, working memory and executive function performance) in adolescents with a first episode of EOP and controls over a 2 year period, using data from the longitudinal child and adolescent first-episode psychosis study (CAFEPS) [56]. Progressive brain changes in specific frontal and parietal volumes in the CAFEPS sample have previously been described, with individuals with EOP presenting with greater loss of GM volume than controls, over the two years after the onset of psychotic symptoms [13,17,18]. Moreover, a significant cognitive impairment in sustained attention, working memory and executive function over time was found in EOP individuals in the CAFEPS sample over the same period of time [14]. Therefore, we aimed to investigate the relationship between these two events in those participants with EOP and controls who underwent baseline and follow-up brain and cognitive assessments in the context of the CAFEPS study. Therefore, we aimed to take these results a step further, to investigate the relationship between these two events in those participants with EOP and controls who underwent baseline and follow-up brain and cognitive assessments using the same previously validated methodology. The study of EOP participants whose brain and cognitive development is still in progress at the time of the first episode provides a unique opportunity to compare their brain and cognitive maturation with that of controls, in search of specific patterns resulting from plausible neurodevelopmental processes occurring at this particular stage. We hypothesized that a progressive loss of frontal and parietal GM volume over the 2 year follow-up will be associated with poorer cognitive development in high-order cognitive functions that have traditionally been associated to fronto-parietal areas, such as sustained attention, working memory and executive function, in adolescents with a first episode of EOP.

## 2. Materials and Methods

### 2.1. Participants

The detailed description of participants, recruitment and complete methodology of the multicenter longitudinal follow-up CAFEPS study in Spain has been reported previously [56]. A total of 110 individuals with a first episode of psychosis and their corresponding 98 matched controls were consecutively recruited in child and adolescent outpatient and inpatient units in 6 hospitals in Spain. The recruitment took place between March 2003 and November 2005, according to the following inclusion and exclusion criteria. Participants were included if they/their legal guardians provided written informed consent and were aged between 7 and 17 years old and presented with a first episode of EOP of less than 6-months duration. Concomitant Axis I disorder, DSM-IV mental retardation, presence of any neurologic or pervasive developmental disorder, substance abuse, pregnancy or history of head trauma with loss of consciousness were exclusion criteria. Substance use was not an exclusion criterion if symptoms persisted 14 days after a negative routine urine drug test at first admission to the clinical centers. Controls were recruited from the same sociodemographic areas through dissemination by the hospital staff, word-of-mouth and advertisements. The study was approved by all the corresponding Institutional Review Boards at each of the participating clinical centers. All participants met MRS safety criteria. For this study purpose, only those participants that completed both baseline and longitudinal cognitive and magnetic resonance imaging (MRI) assessments were included in the analyses. Thus, a subsample of 33 first-episode EOP subjects (mean age 15.82; range [11,12,13,14,15,16,17]) and 47 matched controls (mean age 15.26; range [13,14,15,16,17]) (mean inter-scan interval (SD) = 25.11 [2.48] months) were considered from the original CAFEPS sample (see Figure 1). There were no differences in age, sex or clinical or functional characteristics between those participants with EOP who completed and the ones who did not complete the correspondent baseline and 2 year follow-up MRI and cognitive assessments (i.e., had either no valid baseline and/or follow-up MRI and/or cognitive data—see Appendix A).

### 2.2. Clinical Assessment

Diagnosis or its absence was established at baseline according to DSM-IV criteria, using the Spanish version of the semi-structured diagnostic interview; the Kiddie-Schedule for Affective Disorders and Schizophrenia, Present and Lifetime version (K-SADS-PL) [57,58]; with which accuracy of diagnosis of psychosis was reviewed at the 2 year follow-up. The diagnosis established at follow-up was used for descriptive purposes only, and was grouped into three main diagnostic categories: schizophrenia (N = 13) bipolar disorder (N = 11) and other psychoses (N = 9; including schizoaffective disorder N = 2, depression with psychotic features N = 1, psychosis not otherwise specified N = 6).

The same trained psychiatrist conducted clinical and functional assessments for each EOP participant at both baseline and 2 year visits. Severity of symptoms was assessed using the Positive and Negative Syndrome Scale (PANSS) [59,60]. Prior to recruitment, inter-rater reliability for the PANSS was determined in an independent sample of 10 individuals with psychosis, using the interclass correlation coefficient (ICC), which was always superior to 0.80. Level of general occupational, social and psychosocial functioning was determined using the Children Global Assessment of Functioning (C-GAF) scale [61]. Duration of untreated psychosis was calculated by asking EOP subjects/parents at the first interview about the approximate date of first appearance of positive psychotic symptoms within the first psychotic episode and by reviewing clinical case notes [62]. Neurodevelopmental history (normal or pathological acquisition of psychomotor, language and reading and writing milestones) was also explored by asking the parents at baseline. Chlorpromazine equivalents were used to derive the dosage of antipsychotic treatment at baseline and follow-up, and to calculate the cumulative doses taken during the scan interval [63,64].

### 2.3. Cognitive Assessment

Cognitive function was assessed at baseline and at two year follow-up using a comprehensive neuropsychological battery, including those cognitive functions consistently described as being affected in psychosis [1,2] (i.e., sustained attention, working memory, learning and memory and executive function). Participants completed this cognitive evaluation when considered to be psychopathologically stable (i.e., within 4 weeks after recruitment to the study). Of those, only high-order cognitive measures considered to be related to the frontal [42,47,65,66,67,68,69,70] and parietal regions [71,72] were selected, comprising the following cognitive test and their correspondent derived neuropsychological variables of sustained attention (digits forwards subtest of the Wechsler Adult Intelligence scale, WAIS-III (Wechsler Adult Intelligence Scale, WAIS-III; 1997), time to complete Trail Making Test part A, TMT-A [73], total number of correct items from Words and Colors Stroop Test [74], total number of correct responses and average reaction time from the Continuous Performance Test, CPT [75]); working memory (digits backwards and letter-number sequencing subtests, WAIS-III [76]); and executive function (TMT B/A [73], number of errors, number of perseverative errors and number of categories of the Wisconsin Card Sorting Test, WCST [77], Stroop interference score and total number of correct words from the Verbal Fluency Test, FAS (Benton y Hamsher, 1989) and total number of correct words from the Controlled Oral Word Association Test, COWAT [78]) (See Table 1).

Psychologists that were trained in the use of the cognitive battery conducted all neuropsychological assessments. Inter-rater reliability for administration and scoring of the cognitive scales was determined using the ICC in an independent sample of 10 subjects prior to the baseline assessments (>0.85 for all instruments). Raw test scores were converted to z-scores (mean = 0, standard deviation, SD = ±1) based on the cognitive performance of the control group at baseline. The sample was divided into three age groups to minimize the effect of age and education on cognitive performance (aged 11–14, aged 15–16 and aged 17 in accordance with our previous research in the CAFEPS sample [14,23]. A summary score for each cognitive domain at both baseline and 2 year assessment times, and a measure of change at follow-up (2 year follow-up minus baseline) was calculated based on z-scores. The z-score sign was changed from plus to minus and vice versa for the higher scores, to always reflect better performance as required (i.e., time to complete the TMT-A and TMT-B, WCST errors). Z-scores were also truncated at ±4, to avoid outlying variables.

### 2.4. Brain Imaging

#### 2.4.1. MRI Acquisition and Processing

Participants underwent anatomical brain MRI at baseline and at the 2 year follow-up visit in five different 1.5-T scanners: two Siemens Symphony, two General Electric Signa and one Philips ACS Gyroscan. Each individual was scanned with the same scanner at both baseline and follow-up. Data were collected from each center and processed at one site. MRI scans were acquired in axial orientation for each subject, a T1-weighted 3D gradient echo (voxel size 1 × 1 × 1.5 mm) and a T2-weighted Turbo-Spin-Echo sequence (voxel size 1 × 1 × 3.5 mm). Full details about the acquisition parameters at each site, comparability between machines for this study and the limitations involved in the analysis of multicenter data are provided elsewhere [79]. In keeping with our previous studies using this sample [17,18,80], lobar volumes of GM were obtained by using an automated method based on the Talairach proportional grid system [79,81,82,83,84]. Processed images were visually inspected and manually corrected by a single blind operator. Brain images from a sample of five randomly selected participants were used to test the consistency of the rater prior to conducting the whole segmentation procedure in the images of the study sample. The intra-class correlation coefficients ranged from 0.96 to 0.99 for lobar GM measurements [79].

#### 2.4.2. Segmentation and Regions of Interest (ROI) Definition

MRI images were processed using locally developed software, incorporating a variety of image processing and quantification tools [79,84]. To obtain lobar volume measurements, we used a method for semi-automated segmentation of the brain based on the Talairach proportional grid system, which includes a grid template in which all sectors are assigned to particular regions [81,82]. A two-step procedure was followed [84]: (1) An initial segmentation of cerebral tissues into GM, white matter (WM) and CSF was obtained using Statistical Parametric Mapping 2 (SPM2) routines for multimodal (T1 & T2) segmentation. The SPM algorithm for tissue segmentation includes a multi-modal method to eliminate the effect of radiofrequency field inhomogeneities [83], which was proven to be more robust than single-modality in a multicenter setup [79]. (2) The Talairach grid was built on each edited brain MRI by manually selecting the position of the anterior and posterior commissures (AC, PC), and establishing a third point position in the mid-sagittal plane. The AC-PC line was set in the axial horizontal plane, and the inter-hemispheric plane in the vertical orientation [85]. Our software application automatically finds the outer brain limits in Talairach orientation, and 3D grids are then built for each brain. The Talairach grid, a piecewise linear transformation and a tessellation of the brain into a 3D grid of 1056 cells, represents homologous brain regions across subjects [85]. The ROI measurements were obtained by superimposing the 3D tissue masks corresponding to GM, WM and CSF onto each subject’s Talairach grid, where the regions of interest were defined as sets of Talairach grid cells [79,82,83]. Volumes for each tissue type in Talairach space were measured on the MRI registered to the Talairach space, by summing up the data from the Talairach grid cells associated with each ROI [84].

The validity of this procedure has already been proven suitable for volumetric studies [81,82,84] in our own longitudinal [17,18] and other multicenter studies [79,80,86].

Based on our own previous findings on brain volume changes and cognitive development in this sample [14,17], the ROIs selected and included in the analysis were the frontal and parietal lobes. These fronto-parietal regions were among the brain regions most likely to present volume changes over time in adolescent first-episode subjects [17,18,87,88], and have been associated with high-order cognitive processes, such as sustained attention, working memory and executive function [89,90,91,92,93,94,95,96,97,98,99]. Volumes were obtained in both left and right hemispheres. Intracranial volume (ICV) was derived as the sum of the total GM, WM and CSF volumes, including the cerebellum [17,18]. This data was exported to SPSS, version 25.0 (Chicago, Illinois) for statistical analyses.

The longitudinal change in volume was measured as the difference between follow-up (vol2) and initial volume at baseline (vol1) of the ROI, and described as a percentage calculated as follows [17,18]:Longitudinal Change = ((vol2 − vol1)/vol1) × 100

### 2.5. Statistical Analyses

Normal distribution of variables was examined with the Kolmogorov–Smirnov test. Means and standard deviations (SD) were used to describe continuous variables. Frequencies were used to describe discrete variables. Differences between EOP individuals and controls in demographic, neuropsychological and clinical data were assessed by means of Student’s t-test, Fisher’s test and chi-square tests, according to the type of variable.

In order to examine the relationships between change in frontal and parietal GM volumes and change in cognitive performance in the sustained attention, working memory and executive function domains (composite z-scores), a series of backwards stepwise regression analyses were conducted. These models were built using relative within-subject longitudinal volume change (described above as a percentage) in bilateral frontal and parietal GM volumes, respectively, as independent variables, and changes at follow-up (2 year follow-up minus baseline) in cognitive outcomes as dependent variables. In congruence with our previous studies using this sample, the models included age, months of inter-scan interval and inter-scan ICV change as covariates of no interest. These three covariates were included in the model because of their potential effect on volume data. Assuming that ICV would be constant, inter-scan ICV change was included to remove a potential effect of spurious method variance associated with change from ICV1 to ICV2 [17,18]. In order to increase predictive power and avoid overfitting the model, a backwards stepwise regression model was built for each dependent variable, including one ROI at a time as a predictive variable together with the variables of no interest. These models were built for EOP participants and controls separately.

A final model was built in order to further analyze the predictive value of those independent variables that demonstrated significant associations with the primary variables of interest, entering only those variables, which emerged as significant predictors for each cognitive outcome.

Pearson’s linear correlation coefficients were used to rule out potential effects of medication on brain changes, and of medication and symptom changes on cognitive performance over time by examining their associations.

All statistical analyses were performed using IBM SPSS v25 (Statistical Package for the Social Sciences, IBM Corporation). Statistical significance was set at α < 0.05.

## 3. Results

### 3.1. Sociodemographic and Clinical Characteristics

Individuals with EOP were not significantly different from controls, in terms of age, sex, parental socioeconomic status, years of education, developmental history, race or handedness (see Table 2). There were significant differences between groups for estimated IQ at baseline, consistent with previous reports on this sample [14,23] and for months of inter-scan interval in our sample (see Table 2). The mean duration of the illness, defined as the time between the appearance of their first positive symptoms and their baseline MRI scan, was 3.12 ± 2.75 months for the EOP sample included in the current analyses. Duration of antipsychotic treatment at baseline was 3 ± 1.87 weeks (mean daily dose 277.7 ± 150.4 mg in chlorpromazine equivalents [63,64].

### 3.2. Relationship and Predictive Value of Change in Brain Volume Measures over Change in Cognitive Performance at Two Year Follow-Up

A total of four backwards regression models were built for each cognitive outcome (sustained attention, working memory, executive function) including one ROI for time together with the variables of no interest as predictors, for EOP individuals and controls separately (see Methods). For the sake of conciseness and clarity, only the main significant results are provided in Table 3. Tables showing all the regression models can be seen in Appendix A.

We found that longitudinal change (decrease) in frontal GM volume was positively associated with decreased working memory performance in individuals with EOP (left (B = 0.096, *p* = 0.008); and right (B = 0.089, *p* = 0.015)) as a function of age (see Table 3). In addition, decreased parietal GM volume was positively associated with decreased working memory performance in this group ((left (B = 0.119, *p* = 0.007), right (B = 0.125, *p* = 0.015)) also as a function of age (see Table 3). Specifically, for the first multiple regression model, including change in frontal left GM volume over time as an independent variable, a decrease in frontal left GM volume over time explained a significant amount (18.9%) of the variance of change (decrease) in working memory, when variance for other variables in the final model was accounted for (age 16.4%). In the second multiple regression model, with change in frontal right GM volume over time entered as an independent variable, both change in frontal right GM volume and age variables added statistically significantly to the variance of change in working memory, after adjustment of other predictors in the model. In particular, decreased frontal right GM volume explained 16% of the variance of decrease in working memory performance over time, accounting for variance in age (12%). In the final solution of the third multiple regression model (including change in parietal left GM volume as an independent variable), both change in parietal left GM volume and age variables added statistically significantly to the variance of change in working memory, with decreased parietal left GM volume explaining 19.5% of the variance of decrease working memory performance over time. This was independent of age, which accounted for 17.5% of the variance in this model. The fourth regression model, with change in parietal right GM volume as an independent variable, explained a significant proportion of the variability of change in working memory performance over time. Specifically, a decrease in parietal right GM volume over time explained a significant amount (16%) of the variance of change (decrease) in working memory performance over time, when variance for other variables in this final solution was accounted for (age 14.9%).

A final multiple regression model was built in order to further examine the predictive value of the significant results obtained, regarding the reported relationship between variables of longitudinal change in frontal and parietal GM volume (left and right), age and working memory performance over time. In this model, a particular decrease in frontal left GM volume over the two year follow-up best predicted a significant amount (22.3%) of the variance of decreased working memory performance, together with age which accounted for 18% of the variance of longitudinal performance in this cognitive domain (Table 3).

No significant relationships were found for any of the brain GM volume measures and cognitive variables of sustained attention or executive function performance over time in individuals with EOP. There were no significant relationships between any of the longitudinal change in brain GM volume measures and cognitive variables in the control group.

These results were independent of antipsychotic exposure and symptom changes (see Appendix A).

## 4. Discussion

This is the first study to longitudinally examine the relationship between brain GM changes and high-order cognitive function development in a sample of adolescents with EOP and a sample of controls, with the latter adding to the novelty of our findings. Our results suggest a particular association between decreased GM volume of frontal and parietal lobes and a poorer working memory performance over the first 2 years of psychosis in adolescents with a first episode of EOP as a function of age. In particular, within EOP individuals but not controls, loss of GM volume in frontal and parietal regions and younger age at the time of the FEP were associated with decreased working memory function over time. No such association was found with other high-order cognitive functions (i.e., for sustained attention or executive function) in this sample. Further, specifically frontal left GM loss together with age have shown to be the best predictors of working memory performance over time in adolescents with EOP. No associations were found between any of the measures of brain GM volume change and the cognitive variables in the control group.

These findings are congruent with our previous studies in this sample, on which a specific arrest in cognitive development was observed for working memory function in the larger sample of EOP from which our sample was derived [14]. In particular, although EOP individuals presented with significant cognitive impairment with regards to controls at baseline and at the 2 year follow-up, both controls and EOP individuals improved in all cognitive measures over time, except for EOP working memory [14]. These changes occurred in tandem with a prominent reduction in frontal and parietal GM volume in EOP with regards to controls in this same larger sample, and a decrease was particularly evident for these fronto-parietal regions in the left hemisphere [17,18]. Our study now demonstrates the relationship between these two events as a function of age. Specifically, in the subset of individuals who completed both cognitive and brain imaging measures at baseline and 2 year follow up, our results suggest EOP is characterized by decreased working memory function over time, associated with age and accelerated brain GM volume loss in frontal and parietal structures, two neuroanatomical locations traditionally associated with working memory function [89]. Moreover, a specific reduction of GM brain volume in the frontal left structure was particularly predictive of working memory performance as a function of age.

Impairment in working memory has been previously linked to alterations in brain structure [68] and function [100] in psychosis. In accordance with our findings, previous cross-sectional volumetric studies in FEP have found reduced frontal [19,44,45,53] and temporo-parietal [53] GM volume to be associated with poor working memory function, particularly evident in the left frontal region [44]. More specifically, frontal and temporo-parietal gray matter volume has previously been associated with a cognitive index, including measures of sustained attention (forward digit span test), working memory (backward digit span, a test used in our study) and executive functioning (verbal fluency COWAT test) in adults with FEP [53]. Association between spatial working memory (assessed using the spatial working memory task from the Cambridge Neuropsychological Test Automated Battery, CANTAB) and GM volume in the left frontal regions was also particularly evident in young adults with first-episode schizophrenia [44]. In a different study, lower GM volume in the left dorsolateral prefrontal cortex was found to be specifically related to working memory impairment (assessed using the digits span test, as in our study) in adults in the early course of schizophrenia spectrum disorders [45]. Congruent findings with these FEP volumetric studies could reflect a unique pattern of associations in the early course of psychosis. Variability across neuroimaging analysis methodology or measurement strategies for assessing working memory is another possibility. In this regard, our results are also in accordance with previous morphometric studies in FEP, in which reduced cortical thickness in the frontal [101], frontal, parietal, temporal and cingulate [102] or temporal cortices [103] has been associated with working memory function in adults with psychosis. In detail, the study from Gutiérrez-Galve et al. [101] used measures of working memory span (spatial span task) and working memory manipulation (spatial working memory task) derived from the CANTAB, and found associations for working memory span only with frontal cortical area in FEP. Haring et al. [102] also used the spatial working memory test from the CANTAB in FEP individuals, and found associations with frontal, temporal, cingular and occipital cortical parameters in addition to associations of these brain regions with set-shifting, strategy usage and sustained attention cognitive abilities. Ehrlich et al. [103] used letter-number sequencing as a measure of working memory and found associations with cortical thickness in lateral prefrontal cortex in controls, whereas participants with established schizophrenia demonstrated associations with cortical thickness in a distinct right middle and superior temporal lobe brain region. Opposite to this later finding and consistent with previous studies, we have shown that some brain–cognitive associations tend to be specific to individuals with FEP [32,38,44,53,103,104,105]. However, it is important to consider the previously mentioned methodological differences and sample characteristics when trying to generalize findings across studies. In this regard, our study provides complementary findings of longitudinal/developmental brain volume and high order cognitive function associations using verbal/spatial working memory tasks in adolescents with psychosis compared to controls, which adds to the relevance of our findings.

Normative studies have consistently emphasized the role of working memory in cognitive development. Working memory function is crucial for general intellectual development [106,107] and a major predictor of high-order cognitive function development and school achievement [108]. Working memory ability progressively increases with age, from infancy until late adolescence [109,110,111,112], a time during which the brain undergoes major morphological and functional changes [20,26,27,113,114]. In typical development, fronto-parietal GM volume generally peaks at 11–12 years, whereas gains in working memory are extended up to 15–19 years of age, paving the way for the continuous development of executive function up to the mid-twenties [24]. Sustained attention, a basic underlying cognitive process required to complete any planning activity, is the first one to develop [10,89]. These were the three high-order cognitive domains assessed in our longitudinal study in adolescence. Therefore, the age range of our participants (11 to 17 years old) might represent a particularly sensitive period for the development of GM in the brain, and of cognitive functions, such as working memory (a cognitive milestone of this particular stage), in which the onset of psychosis might precipitate accelerated fronto-parietal brain volume changes, which in turn alter age-dependent working memory performance during the first two years after the first episode. The influence of age in cognitive [28] and brain abnormalities in psychosis had already been outlined by previous studies in our group [115]. In accordance with this notion, an accelerated loss of fronto-parietal GM volume was observed in our EOP sample (particularly relevant in the left frontal region), associated with working memory dysfunction over the first two years after the first psychotic episode as a function of age, indicating poorer working memory performance over time in younger participants with EOP. Interestingly, no such relationship was found for sustained attention or executive function, highlighting the relevance of these associations for this age range. In our study, these are associations specific of EOP, as they were not observed for adolescents in the control group. In other words, results partially confirmed our hypothesis that reduced fronto-parietal GM volume is associated with high-order cognitive function in EOP. In contrast to associations with working memory function, we did not find an association between fronto-parietal GM volumes and sustained attention or executive function in our sample. Our findings suggest that fronto-parietal brain GM loss (particularly in the left frontal lobe) may involve disruption of working memory development in adolescent psychosis, and thus be considered as a potential biomarker. The neurodevelopmental hypothesis emerges as a plausible explanation of the etiopathological mechanisms that might underlie cognitive development in our EOP sample at this particular point.

Limitations of the present study include the relatively small sample size, which prevented us from examining brain and cognitive changes in specific diagnostic subgroups, and may lead to unrepresentative groups of participants in comparison to the main study from which this sample was derived [56]. On that account, no differences were observed between those individuals with EOP who completed both cognitive and MRI assessment at baseline and 2 year follow-up and those who did not complete them. Second, the short follow-up of 2 years might not be fully representative of a broader pattern of cognitive development and brain changes in this stage of adolescence. We are currently conducting 10 year follow-up assessments for the same sample of individuals with EOP and controls, in order to further explore the trajectories of cognitive and brain development and its relationships over time in these subjects. Third, even though we found no effect of symptom severity or antipsychotic medication, the absence of a medication-free comparison group prevents the assessment of the impact that antipsychotic medication may have had on our findings. Fourth, our analysis protocol is restricted to frontal and parietal brain lobes and high-order broad cognitive domains, such as sustained attention, working memory and executive function, which constrains our findings to other cognitive domains/specific cognitive processes or to subcortical structures or larger-scale brain changes. Fifth, a potential interference of practice effects in neuropsychological testing might be another limitation. This interference is an inherent methodological limitation in longitudinal neuropsychological studies. It seems to be higher in repeated measures in short periods of time, and it seems common to all neuropsychological tasks [116]. Last, this was an exploratory study. Thus, corrections for multiple comparisons were not done, which could be perceived as a limitation. However, there is no current agreement among statisticians about the need to make p-value adjustments while analyzing data [117,118,119,120,121]. Since multiple testing adjustments control false positives at the potential expense of false negatives, we chose to report all comparisons and p-values and regard our findings as tentative [117,119,122] Future studies should evaluate individual components of the high-order cognitive functions evaluated, specifically of working memory performance and adopt a whole-brain and fine-grained interdisciplinary approach that incorporates multimodal imaging, biochemical, genetic and functional measures. This is in order to better understand the neural bases of cognitive function in psychosis, particularly the complex interplay between the observed longitudinal brain changes and cognitive outcomes, in a field where there is still a lack of effective treatments.

Strengths include the short duration of the illness and of the antipsychotic treatment at baseline. The homogeneity of the sample, in terms of age and education, limits the impact of sociodemographic factors and maturational processes in our findings. The follow-up duration was established carefully to be the same for all participants, reducing the confounding factors that could be accounting for longitudinal changes. This study may also have important clinical implications for the development and implementation of both cognitive training programs and novel pharmacological treatments. Adolescence is a time of continued brain and cognitive development. It thus represents a critical time window of neural plasticity when different brain regions and superior cognitive functions are still maturing. Considering our findings, the implementation of effective cognitive and pharmacological strategies targeting brain and cognitive changes at the time of the first episode are thus of utmost importance, to mitigate the effects of psychosis on neurodevelopmental processes that might underlie persistent cognitive deficits in psychosis.

## 5. Conclusions

Our results highlight the relationship between brain volume changes in fronto-parietal regions, and the development of working memory function in adolescents with EOP as a function of age. These two events co-occur in a crucial time of adolescence that are particularly sensitive to working memory and frontal brain volume changes, which might explain the observed “disruption” in these neurodevelopmental processes in adolescents within our age range. A reduction in frontal left GM volume in particular can be considered as a potential predictive biomarker underlying the observed working memory impairment. Thus, our findings may guide the development of new therapeutic approaches based on the neurodevelopmental processes (brain and functional/cognitive changes) occurring at the time of the first episode in EOP. Boosting changes in working memory capacity might stimulate developmental changes in high-order cognitive function, which might lead to better clinical and functional outcomes in this population [123].

## Figures and Tables

**Figure 1 jcm-10-03929-f001:**
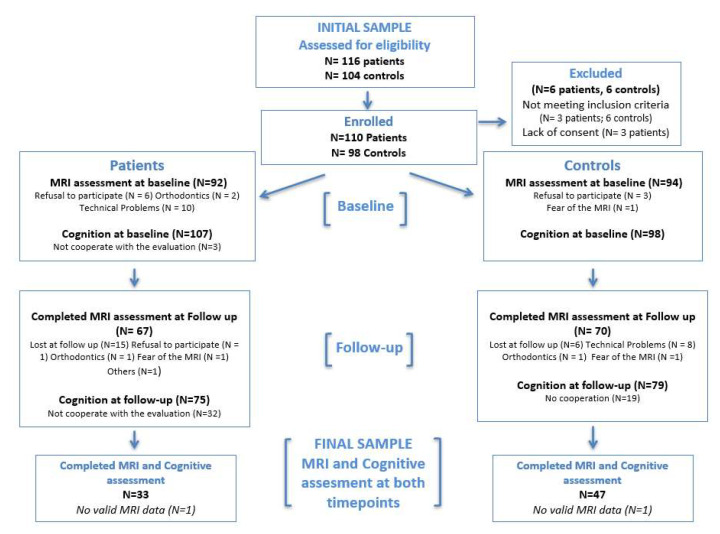
Flowchart for completers and non-completers. Non-completers are participants for whom we do not have a baseline or follow-up cognitive or magnetic resonance imaging data, or from whom the data were not good enough to be used for the study.

**Table 1 jcm-10-03929-t001:** Neuropsychological test and variables used to evaluate cognitive function performance at baseline and at two year follow-up.

Cognitive Domain	Neuropsychological Variable
Sustained Attention	WAIS-III Digits Forward
	Time to complete TMT-A
	Number of correct items Stroop 1 words and 2 colours
	Number of correct responses CPT
	Average reaction time CPT
Working Memory	WAIS-III Digits Backward
	WAIS-III Number–Letter Sequencing
Executive Function	TMT Derived Score TMTB—TMTA
	Number of words FAS
	Number of words COWAT
	Stroop Interference score
	WCST number of perseverative errors
	WCST number of errors
	WCST number of categories

WAIS-III, Wechsler Adult Intelligence Scale, 3rd Edition; TMT-A, Trail Making Test, Part A; TMT-B, Trail Making Test, Part B; CPT, Conners’ Continuous Performance Test; TAVEC, Spanish version of the California Verbal Learning Test; FAS, Verbal fluency test; COWAT, Control Oral Word Association Test (Category Test); WCST, Wisconsin Card Sorting Test.

**Table 2 jcm-10-03929-t002:** Sociodemographic and clinical characteristics of EOP individuals and controls.

	EOP		Controls		Test ^a^	*p*-Value
	*n* = 33		*n* = 47			
	Mean	DS	Mean	DS		
**Age (Years) [range]**	15.82 [11,12,13,14,15,16,17]	1.467	15.26 [13,14,15,16,17]	1.343	t = 1.776	*p* = 0.080
**Sex (Female/Male)**	14/19		17/30		X2 = 0.319	*p* = 0.572
**Education (Years)**	8.885	1.805	8.77	1.371	t = 0.232	*p* = 0.817
**Parental Socioeconomic Status ^b^ (1/2/3/4/5)**	7/10/7/5/4/		3/14/13/3/14		X2 = 7.915	*p* = 0.095
**Race/Ethnicity (Caucasian/Other)**	32/1		44/3		X2 = 0.459	*p* = 0.639 Fisher’s exact test
**Diagnosis at follow-up ^c^ (SZ/BP/Others)**	13/11/09		-	-	-	-
**Handedness (Right/Left/Mixed)**	27/6/00		42/3/2		X2 = 3.931	*p* = 0.140
**Psychomotor Development (No. Normal/Pathology)**	31/2		44/3		X2 = 0.003	*p* = 0.953 Fisher’s exact test
**Language Development (No. Normal/Pathology)**	27/6		42/5		X2 = 0.930	*p* = 0.347 Fisher’s exact test
**Reading and Writing Development (No. Normal/Pathology)**	29/4		44/3		X2 = 0.800	*p* = 0.439 Fisher’s exact test
**Estimated IQ (Baseline)**	81.21	16.435	104.32	14.223	t = −5.998	*p* = 0.000
**PANSS Negative (Baseline)**	18.82	8.921	-	-	-	-
**PANSS Positive (Baseline)**	24.97	5.987	-	-	-	-
**GAF (Baseline)**	34.64	15.56	-	-	-	-
**Duration of untreated psychosis (Days)**	53.58	52.431	-	-	-	-
**MRI Between-scan follow-up period (Months)**	24.45	2.152	25.57	2.483	F = 0.33	*p* = 0.039
**Duration of antipsychotic treatment at baseline MRI (Weeks)**	3	1.87	-	-	-	-
**Cumulative antipsychotic dosage (mg) Chlorpromazine equivalents ^d^**	149.369, 2	104.428, 42	-	-	-	-
**Antipsychotic treatment at baseline (No.) (Olanzapine/Risperidone/Quetiapine/Ziprasidone) ^e^**	9/18/8/2		-	-	-	-
**Antidepressive Treatment at baseline (No.) (Paroxetine/Venlafaxine/Fluoxetine)**	1/2/02		-	-	-	-
**Mood Stabilizers at baseline (No.) (Lithium/Oxcarbazepine/Valproate)**	3/1/01		-	-	-	-

Abbreviations: GAF, Children’s Global Assessment of Functioning; MRI, magnetic resonance imaging; PANSS, Positive and Negative Syndrome Scale. IQ = intellectual quotient. [^a^] Two sample *t*-test was used for comparisons between quantitative measures, and Fisher’s exact or Pearson’s chi-square test was used for comparisons between qualitative measures (*p* < 0.05). [^b^] Parental socioeconomic status assessed with the Hollingshead Scale (ranging from 1 to 5) (Hollingshead and Redlich 1954). A rating of 5 corresponds to the highest socioeconomic status and a rating of 1 to the lowest. [^c^] Diagnosis: SZ, schizophrenia; BP, bipolar disorder; Others (schizoaffective disorder, depression with psychotic features and psychosis not otherwise specified) (see Section 2 “Methods”). [^d^] Chlorpromazine equivalents were used to derive the dosage of antipsychotic treatment at baseline and follow-up, and to calculate the cumulative doses taken during the scan interval. [^e^] Patients were polymedicated, increasing the sample size for medication. At baseline four patients were taking two antipsychotics.

**Table 3 jcm-10-03929-t003:** Associations and predictive value of change in brain volume measures over change in cognitive performance at two year follow-up in EOP (final prediction models for significant backwards regression analyses are displayed—see Methods).

DV *: CHANGE IN WORKING MEMORY OVER TIME					
**PREDICTORS:** Change in Frontal Left GM Volume, age, months of inter-scan interval and interscan ICV change.					
**Model Summary**	**R**	**R^2^**	**Adjusted R^2^**	**Standard Error of the Estimate**	
	0.550	0.303	0.256	1.039	
**Model 4**	**B**	**Standard Error**	**Beta**	**t**	***p***
**Change in Frontal Left GM Volume**	0.096	0.034	0.441	2.854	**0.008**
**Age**	0.027	0.010	0.412	2.666	**0.012**
**PREDICTORS:** Change in Frontal Right GM Volume, age, months of inter-scan interval and interscan ICV change.					
**Model Summary**	**R**	**R^2^**	**Adjusted R^2^**	**Standard Error of the Estimate**	
	0.523	0.274	0.225	1.061	
**Model 4**	**B**	**Standard Error**	**Beta**	**t**	***p***
**Change in Frontal Right GM Volume**	0.089	0.035	0.400	2.570	**0.015**
**Age**	0.023	0.010	0.347	2.230	**0.033**
**PREDICTORS:** Change in Parietal Left GM Volume, age, months of inter-scan interval and interscan ICV change.					
**Model Summary**	**R**	**R^2^**	**Adjusted R^2^**	**Standard Error of the Estimate**	
	0.556	0.309	0.263	1.035	
**Model 4**	**B**	**Standard Error**	**Beta**	**t**	***p***
**Change in Parietal Left GM Volume**	0.119	0.041	0.451	2.910	**0.007**
**Age**	0.028	0.010	0.426	2.752	**0.010**
**PREDICTORS:** Change in Parietal Right GM Volume, age, months of inter-scan interval and interscan ICV change.					
**Model Summary**	**R**	**R^2^**	**Adjusted R^2^**	**Standard Error of the Estimate**	
	0.523	0.274	0.225	1.061	
**Model 4**	**B**	**Standard Error**	**Beta**	**t**	***p***
**Change in Parietal Right GM Volume**	0.125	0.049	0.403	2.571	**0.015**
**Age**	0.025	0.010	0.389	2.480	**0.019**
**FINAL MODEL. PREDICTORS:** Change in Frontal Left GM Volume, Change in Frontal Right GM Volume, Change in Parietal Left GM Volume, Change in Parietal Right GM Volume, age.					
**Model Summary**	**R**	**R^2^**	**Adjusted R^2^**	**Standard Error of the Estimate**	
	0.554	0.318	0.272	1.028	
**Model 4**	**B**	**Standard Error**	**Beta**	**t**	***p***
**Change in Frontal Left GM Volume**	0.107	0.034	0.490	3.131	**0.004**
**Age**	0.362	0.129	0.441	2.815	**0.009**

* Dependent variable (DV), gray matter (GM).

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
