# Peer review of "Fronto-Parietal Gray Matter Volume Loss Is Associated with Decreased Working Memory Performance in Adolescents with a First Episode of Psychosis"

_jcm, 2021, doi:10.3390/jcm10173929_

Round 1

Reviewer 1 Report

In the present paper Rapado-Castro et al. investigate the relationship between GM volume loss and cognitive impairment in specific domains in adolescent subject affected by EOP. The paper is well written and presented, but some concerns should be addressed:

  • Recent neuroimaging studies are focused on extracting detailed analyses and parameters describing the brains, characterizing it beyond the sole gray matter volume. The authors should perform the anaylses leveraging neuroimaging analyses pipeline or comment the choice of relying only on Talairach alignment and GM lobar volume measurement instead of state-of-the-art analyses pipeline for gray matter volumetric analyses
  • The authors should include all the mentioned regression models, including non significant results, as supplementary data. Same applies for results not shown regarding antypsychotic exposure and symptom change analyses.

Author Response

REVIEWER #1.

Comment 1. “In the present paper Rapado-Castro et al. investigate the relationship between GM volume loss and cognitive impairment in specific domains in adolescent subject affected by EOP. The paper is well written and presented, but some concerns should be addressed:

Recent neuroimaging studies are focused on extracting detailed analyses and parameters describing the brains, characterizing it beyond the sole gray matter volume. The authors should perform the analyses leveraging neuroimaging analyses pipeline or comment the choice of relying only on Talairach alignment and GM lobar volume measurement instead of state-of-the-art analyses pipeline for gray matter volumetric analyses”

Response 1. We thank the reviewer for the positive comments. We are aware of recent neuroimaging studies using more detailed and comprehensive analyses and agree on the need of characterising the brain beyond the sole gray matter volume, and so we considered that as a limitation (see page 18). A whole-brain approach in particular appears more advantageous, however this was an exploratory study, based on previous findings on this sample that had previously been characterised using this same validated methodology [1, 2]. Therefore, in keeping with our previous studies using this sample, were progressive brain changes were found in specific frontal and parietal volumes (with individuals with EOP presenting with greater loss of GM volume than controls over two-years after the onset of psychotic symptoms [1, 2]); and the presence of a significant cognitive impairment in sustained attention, working memory and executive function over time was established (in the same group of EOP individuals over the same period of time [3]); we aimed now to take these results a step further to investigate the relationship between these two events in those participants with EOP and controls who underwent baseline and follow-up brain and cognitive assessments using the same methodology (see methods section). Further, the MRI voxel-resolution of our subjects was 1x1x1.5mm, which additionally prevented from using state-of-the art pipelines such as Freesurfer as these work improperly with slice thickness >1.2mm. These particular aspects have now been added to the introduction (see pages 3 to 4) and emphasized in the correspondent methods section (see page 7)

Moreover, following the reviewers suggestion we stated in the discussion, limitations section; “Future studies should (…) adopt a whole-brain interdisciplinary approach that incorporates multimodal imaging, biochemical, genetic, and functional measures in order to better understand the neural bases of cognitive function in psychosis, particularly the complex interplay between the observed longitudinal brain changes and cognitive outcomes in a field where there is still a lack of effective treatments” (see pp. 18)

Comment 2. The authors should include all the mentioned regression models, including non significant results, as supplementary data. Same applies for results not shown regarding antypsychotic exposure and symptom change analyses.”

Response 2. We thank the reviewer for raising this point. As suggested by the reviewer, we are now providing all the mentioned regression models including non-significant results as supplementary data (see table 3 for significant regression results and supplementary table 2 for non-significant regression results). According to the reviewer’s suggestion, results regarding antipsychotic exposure and symptom change analyses have also now been presented as supplementary data (see supplementary table 3).

References: Response to Comments Reviewer #1.

  1. Arango, C., et al., Progressive brain changes in children and adolescents with first-episode psychosis. Arch Gen Psychiatry, 2012. 69(1): p. 16-26.
  2. Rapado-Castro, M., et al., Gender effects on brain changes in early-onset psychosis. Eur Child Adolesc Psychiatry, 2015. 24(10): p. 1193-205.
  3. Bombin, I., et al., Neuropsychological evidence for abnormal neurodevelopment associated with early-onset psychoses. Psychol Med, 2013. 43(4): p. 757-68.

Reviewer 2 Report

This is an interesting paper overall that links longitudinal gray matter volume changes in frontal and parietal regions to working memory function in an age-related fashion in EOP.   In general, the authors should make sure all typos are fixed, especially when decimal points are expected instead of commas; for example, line 351 and Table 2    Where are letters d and e in table 2?   Figure 1, unclear how the numbers changed from follow up to the final set of boxes   Results concept: Since brain volume on MRI is known to be the combination of surface area and cortical thickness, do the authors have measures of these two features for the ROIs in question? It would provide more detailed information than volume alone.   Table 3:  Is there a different table that shows decreased parietal GM volume association with WM and age? I don't see it in table 3, even though it is cited in the text   Line 426: It is important to show the results that antipsychotic exposure and symptom changes did not confound the results.   Line 447: Is arrest the proper term or is it more of a disruption?

Author Response

REVIEWER #2.

Comment 1. “This is an interesting paper overall that links longitudinal gray matter volume changes in frontal and parietal regions to working memory function in an age-related fashion in EOP.”  

Response 1. We thank the reviewer for the positive remark.

Comment 2. “In general, the authors should make sure all typos are fixed, especially when decimal points are expected instead of commas; for example, line 351 and Table 2    Where are letters d and e in table 2?”.

Response 2. We thank the reviewer for bringing this oversight to our attention. We have now amended all typos both in the text and corresponding tables, particularly decimals points and subscripts. In this regard, letters “d” and “e” in table 2 correspond to “Cumulative antipsychotic dosage (mg) Chlorpromazine equivalents” and “Antipsychotic treatment at baseline (No.) (Olanzapine/Risperidone/Quetiapine/ Ziprasidone)” respectively (See table 2). We hope the manuscript reads more clearly now.

Comment 3. “Results concept: Since brain volume on MRI is known to be the combination of surface area and cortical thickness, do the authors have measures of these two features for the ROIs in question? It would provide more detailed information than volume alone?”.

Response 3. We thank the reviewer for this comment. We agree with the reviewer that surface area and cortical thickness measures would provide more detailed information than volume alone, and so we considered that as future line for investigation (see page 18). In the current work and keeping with our previous studies using this sample, were progressive brain changes were found in specific frontal and parietal brain volumes [1]; and the presence of a significant cognitive impairment in specific cognitive domains over time was established (in the same group of EOP individuals over the same period of time [1]); we aimed now to take these previous results a step further to investigate the relationship between these two events using the same brain and cognitive measures (see methods section). As we agree with the reviewer in the need to conduct such fine-grained analyses, we have stated in the discussion, limitations section; “Future studies should (…) adopt a whole-brain and fine-grained interdisciplinary approach that incorporates multimodal imaging, biochemical, genetic, and functional measures in order to better understand the neural bases of cognitive function in psychosis, particularly the complex interplay between the observed longitudinal brain changes and cognitive outcomes in a field where there is still a lack of effective treatments” (see pp. 18)

Comment 4. “Table 3:  Is there a different table that shows decreased parietal GM volume association with WM and age? I don't see it in table 3, even though it is cited in the text”

Response 4. Again, we thank the reviewer for bringing this oversight to our attention. Table 3 was incomplete in the previous version of the manuscript and has now been amended to include parietal GM volume associations with WM and age (See table 3).

Comment 5. “Line 426: It is important to show the results that antipsychotic exposure and symptom changes did not confound the results”

Response 5. We thank the reviewer for making this point. According to the reviewer’s suggestion, we are now presenting all the results regarding antipsychotic exposure and symptom change analyses as well as all the mentioned regression models including non-significant results as supplementary data (see response 2 to Reviewer 1 and supplementary tables 2 and 3).

Comment 6. “Line 447: Is arrest the proper term or is it more of a disruption?”

Response 6. We thank and agree with the reviewer the use of the term disruption may be more accurate. As such, it has been changed throughout the manuscript.

References: Response to Comments Reviewer #2.

  1. Arango, C., et al., Progressive brain changes in children and adolescents with first-episode psychosis. Arch Gen Psychiatry, 2012. 69(1): p. 16-26.
  2. Bombin, I., et al., Neuropsychological evidence for abnormal neurodevelopment associated with early-onset psychoses. Psychol Med, 2013. 43(4): p. 757-68.

Reviewer 3 Report

This article analyzes the relationship between gray matter volume changes and the results of psychological and cognitive neurological function scale measurements. Such longitudinal research is very meaningful for the understanding of the interaction between brain structure and function.

  1. In the article by Xiao‐Xiao Shan et al. their results provide evidence of increased frontal GMV in prodromal psychosis individuals, however, your study demonstrated the decreased GMV. Please explain the reasons leading to these contradictory conclusions.

Xiao‐Xiao Shan, et al. Increased frontal gray matter volume in individuals with prodromal psychosis. CNS Neurosci Ther. 2019 Sep; 25(9): 987–994. Published online 2019 May 25. doi: 10.1111/cns.13143

  1. Line 112-114, “Two studies, in particular, have investigated the GM volume-cognition relationship in young adults with FEP.” Please add references.

  1. In part Regions of Interest (ROI) definition, could you please specify which software you have applied, and which brain template used for cortical parcellation? Has the brain structure been normalized?

  1. In line 302, what do vol 2 and vol 1 stand for? Is the following analysis related to absolute value?

  1. It is recommended to provide an analysis of GMV differences between different types of psychological diseases.

  1. Regarding the comparison between baseline and follow-up MRI, some patients participated twice, and some participated only once. For patients who have only had one MRI scan, are their data not used in the subsequent analysis?

  1. How did the authors define the segmentation standard between gray matter and white matter? If it was defined by the original grayscale or gray matter intensity, it is not consistent between individuals nor between different scanning sessions; how did the authors normalize the grayscale or gray matter intensity?

  1. The authors used different scales for measuring psychological and cognitive functions. Are there any differences between the results of these scales? Are there differences in the relationship between the results of different scales and GMV?

Author Response

REVIEWER #3.

Comment 1. “This article analyzes the relationship between gray matter volume changes and the results of psychological and cognitive neurological function scale measurements. Such longitudinal research is very meaningful for the understanding of the interaction between brain structure and function”.

Response 1. We thank the reviewer for the positive remarks

Comment 2. “In the article by XiaoXiao Shan et al.[1] their results provide evidence of increased frontal GMV in prodromal psychosis individuals, however, your study demonstrated the decreased GMV. Please explain the reasons leading to these contradictory conclusions.

  • XiaoXiao Shan, et al. Increased frontal gray matter volume in individuals with prodromal psychosis. CNS Neurosci Ther. 2019 Sep; 25(9): 987–994. Published online 2019 May 25. doi: 10.1111/cns.13143”

Response 2. We thank the reviewer for raising this concern. We agree with the reviewer the evidence for neuroimaging-cognition studies is somewhat mixed, particularly when examining brain abnormalities in individuals with prodromal psychosis, where gray matter volume has previously been reported as increased in regions of the frontal lobes as found in the article by Xiao‐Xiao Shan et al.[1]. As these same authors noted in their study [4], on the discussion section, pp. 990, "Our findings of increased GMV in the right IFG and right rectus gyrus in prodromal individuals are inconsistent with our hypothesis and the results of most previous high-risk studies, which showed decreased volume in the fronto-temporal regions in the prodromal individuals (…). The present results are also inconsistent with the results of previous Asian high-risk studies". These inconsistencies might be due to variability across cognitive and neuroimaging measurement strategies, antipsychotic treatment, participants heterogeneity, age, illness stage and duration of untreated symptoms. With regards to the differences with our study in particular, recent reviews on neuroimaging across early-stage psychosis have shown the severity and degree of neural abnormalities in prodromal stages are significantly less than that observed later in the initial onset of psychosis or more established illness stages [8]. In this line, our participants are first episode psychoses, not prodromal, and were younger (mean age in our sample was 15.82 ± 1.47 range [11-17]) than participants on Xiao‐Xiao Shan, et al. 2019 study (where mean age was 22.0 ± 5.25 range [18-44]). In addition, Xiao‐Xiao Shan, et al. 2019 investigated associations of GM volume and cognition at a single point, while ours was a longitudinal study. Thus, the age range of our participants [11 to 17 years old] might represent a particularly sensitive period for the development of GM in the brain and of cognitive functions such as working memory (a cognitive milestone of this particular stage), on which the onset of psychosis might precipitate accelerated fronto-parietal brain volume changes (see pp. 17). Therefore, the differential abnormal development of the brain two years following the onset of psychosis in individuals at a particular neurodevelopmental window such as early adolescence may be an explanation for the differences between our findings [8]. Besides, methodological differences in neuroimaging and cognitive measurement strategies may have contributed to the differences in the results obtained.

Comment 3. Line 112-114, “Two studies, in particular, have investigated the GM volume-cognition relationship in young adults with FEP.” Please add references”.

Response 3. We thank the reviewer for bringing this oversight to our attention. References for the two mentioned studies have now been added in pp. 3.

Comment 4. In part Regions of Interest (ROI) definition, could you please specify which software you have applied, and which brain template used for cortical parcellation? Has the brain structure been normalized?”

Response 4. We thank the reviewer for this comment. For ROI definition we used the software from Andreasen et al. (1996) [2], which includes a Talairach grid template in which all sectors have been assigned to particular regions (lobes). Images were linearly transformed to match the Talairach grid template following an established, manual procedure (selecting the position of the anterior and posterior commissures (AC, PC) and establishing a third point position in the mid-sagittal plane. The AC-PC line was set in the axial horizontal plane, and the inter-hemispheric plane in the vertical orientation [3]. We are not entirely sure what the reviewer refers to with "normalization". No brain deformation (i.e. non-linear transformation) took place. Nevertheless, brain structure was quantified in 'Talairach space' and not in 'native image space'. Given that these only differ by a linear transformation the correlation between measures from both spaces will show high correlation.

The methods section has now been expanded to include this information (see section 2.4.2. Segmentation and Regions of Interest (ROI) definition, pages 7 to 8).

Comment 5. In line 302, what do vol 2 and vol 1 stand for? Is the following analysis related to absolute value?”

Response 5. The acronyms vol 2 and vol 1 stand for GM brain volume at the two-years follow up (vol 2) and initial volume at baseline (vol 1). This has now been added to the correspondent text for further clarification (see pp. 8). As stated on methods section the following analyses are related to absolute time 2 (two-years follow up) minus time 1 (baseline) values for cognitive performance and as the difference between follow-up and initial brain volume at baseline of each ROI described as a percentage (see pp. 9).

Comment 6. It is recommended to provide an analysis of GMV differences between different types of psychological diseases”.

Response 6. Thank you for this accurate suggestion. We agree with the reviewer the analyses of the differences between different types of diagnosis would be highly recommended, however the small sample sizes (N=13 for schizophrenia, N=11 for bipolar disorders, N=9 for other psychoses) prevented us from conducting these analyses. We considered the requirements for regression analyses and the danger of obtaining unstable estimates of parameters in multiple regression models with small sample sizes, and opted for the main group of early onset psychosis, EOP analysis. Again, we agree with the reviewer that individual diagnoses might be contributing to the overall observed effect and so we considered that as a limitation (see page 18)

Comment 7. Regarding the comparison between baseline and follow-up MRI, some patients participated twice, and some participated only once. For patients who have only had one MRI scan, are their data not used in the subsequent analysis?”

Response 7. We thank the reviewer for this important remark. As stated in the methods section, for this study purpose, only those participants that completed both baseline and longitudinal cognitive and magnetic resonance imaging (MRI) assessments were included in the analyses. Thus individuals who had only one MRI scan were excluded from the current study (See pp. 4 and figure 1).

Comment 8. How did the authors define the segmentation standard between gray matter and white matter? If it was defined by the original grayscale or gray matter intensity, it is not consistent between individuals nor between different scanning sessions; how did the authors normalize the grayscale or gray matter intensity?”

Response 8. We thank the reviewer for this comment. For segmentation of each scan into gray/white/CSF we used the SPM2 software with default settings [4]. This software is widely used for segmenting T1 images into tissue types (and CSF). Images were entered into the SPM2 segmentation pipeline in native image space. The resulting segmentations were also in native space. The linear transformation to Talairach was then applied to each individual segmentation. This enabled quantification of the amount of tissue type per lobe using the Talairach grid. 

This information has now been added to the manuscript in the methods section (see section 2.4.2. Segmentation and Regions of Interest (ROI) definition in pages 7 to 8).

Comment 9. The authors used different scales for measuring psychological and cognitive functions. Are there any differences between the results of these scales? Are there differences in the relationship between the results of different scales and GMV?”

Response 9. We thank the reviewer for this important remark. We agree with the reviewer that future studies should evaluate individual components (i.e. specific cognitive subtest) of the high-order cognitive functions evaluated, specifically of working memory performance, and thus we considered that as a limitation (see page 18). Based on our own previous findings on brain volume changes and cognitive development in this sample [5, 6], the aim of the current study was to investigate the relationship between those three a priori defined ROIs that have previously demonstrated to differ between patients and controls in this sample (obtained in both left and right hemispheres [5, 6]) and the cognitive domains that have also shown differences [7] and have been consistently associated to those ROIS (i.e. sustained attention, working memory and executive function). As stated in the methods section, in order to increase predictive power and avoid overfitting the model, a backwards-stepwise regression model was built for each dependent variable (one of the three cognitive domains) including one ROI at a time as a predictive variable together with the variables of no interest. These models were built for EOP participants and controls separately, resulting in a total of 12 regression analyses per group. If we were to include specific subtest as dependent variables (the 14 of them), this would result in a total of 56 regression analyses per group, potentially leading to a number of false positives, specially considering our relative small sample size.

Again, as this was an exploratory study, we considered the requirements for regression analyses and the danger of obtaining a number of false positives, and we choose to examine and report only the main cognitive domain analyses and regard our findings as tentative. As we agree with the reviewer that individual subtest might be contributing to the overall observed effect we noted that as an important objective for future investigation (see page 18).

References: Response to Comments Reviewer #3.

  1. Shan, X.X., et al., Increased frontal gray matter volume in individuals with prodromal psychosis. CNS Neurosci Ther, 2019. 25(9): p. 987-994.
  2. Andreasen, N.C., et al., Automatic atlas-based volume estimation of human brain regions from MR images. J Comput Assist Tomogr, 1996. 20(1): p. 98-106.
  3. Talairach, J.a.T., P., Co-planar stereotaxic atlas of the human brain: 3-Dimensional proportional system: An approach to cerebral imaging. 1988, New York.: Thieme Medical Publishers.
  4. Ashburner, J. and K. Friston, Multimodal image coregistration and partitioning--a unified framework. Neuroimage, 1997. 6(3): p. 209-17.
  5. Arango, C., et al., Progressive brain changes in children and adolescents with first-episode psychosis. Arch Gen Psychiatry, 2012. 69(1): p. 16-26.
  6. Bartholomeusz, C.F., et al., Structural neuroimaging across early-stage psychosis: Aberrations in neurobiological trajectories and implications for the staging model. Aust N Z J Psychiatry, 2017. 51(5): p. 455-476.
  7. Bombin, I., et al., Neuropsychological evidence for abnormal neurodevelopment associated with early-onset psychoses. Psychol Med, 2013. 43(4): p. 757-68.

Round 2

Reviewer 3 Report

Thank you for your response. 

There is a key ambiguous point needs to be clarified.

Major point. In the article “Our software application automatically finds the outer brain limits in Talairach orientation, and 3D grids are then built for each brain. The Talairach grid, a piecewise linear transformation and a tessellation of the brain into a 3D grid of 1,056 cells, represents homologous brain regions across subjects [85]. The ROI measurements were obtained by superimposing the 3D tissue masks corresponding to GM, WM, and CSF onto each subject's Talairach grid, where the regions of interest were defined as sets of Talairach grid cells [79, 82, 83]. Volumes for each tissue type in Talairach space were measured on this MRI by summing up the data from the Talairach grid cells associated with each ROI [84].”

  • “The Talairach grid, a piecewise linear transformation and a tessellation of the brain into a 3D grid of 1,056 cells, represents homologous brain regions across subjects [85]”. Did the author register the individual brain into Talairach Space? The measurement should be taken before normalization, otherwise, the transformation process will add an extra interfering factor to the analysis,whether it is linear or non-linear transformation. There will be different algorithms used in the transformation, such as interpolation, which will become an interference factor in the analysis. Do the authors have theoretical or literature support for this in their methodology? Segmentations should be modulated by scaling with the amount of volume changes due to spatial registration so that the total volume of gray matter in the registered image remains the same as it would be in the original images. 
  • “Volumes for each tissue type in Talairach space were measured on this MRI by summing up the data from the Talairach grid cells associated with each ROI [84].” Which image does “this MRI” indicate? Is it the MRI registered to the Talairach Space?
  •  

Author Response

25 August 2021

Dear Editors,

Re: Reviewed version of manuscript ID jcm-1302258

We would like to thank you for giving us the opportunity to submit a new version of the revised manuscript entitled Fronto-Parietal Gray Matter Volume Loss Is Associated With Decreased Working Memory Performance In Adolescents With A First Episode Of Psychosis”: Rapado-Castro et al. addressing the comments made by the reviewers.

As requested, we have addressed the comments point by point on the following page and have indicated changes in the manuscript using the highlighted text function.

We believe this study is very well suited to this particular Special Issue in the Journal of Clinical Medicine and the findings would be of high interest to your readership.

The content of the article has never been published before and is not under consideration for publication elsewhere. We confirm that the relevant codes of ethics and research ethics were upheld through the conduct of the research, including the assignment of authorship (i.e., based on meritorious scientific contribution). All authors have contributed significantly to the paper and are in total agreement with the content of the manuscript.

The corresponding author is:

Dr. Marta Rapado-Castro

Department of Child and Adolescent Psychiatry,

Institute of Psychiatry and Mental Health,

Hospital General Universitario Gregorio Marañón

C/Ibiza 43, 28009,

Madrid, Spain

Phone: +34 91 426 50 05

Fax: +34 91 426 50 04

We look forward to hearing from you regarding a decision. Thank you again for considering our manuscript for publication in the Special Issue entitled “New Opportunities and Challenges of Early Psychosis”, Journal of Clinical Medicine.

Yours sincerely,

Marta Rapado-Castro, PhD
